# Rashba splitting in organic–inorganic lead–halide perovskites revealed through two-photon absorption spectroscopy

Evan Lafalce [1], Eric Amerling[2], Zhi-Gang Yu [3], Peter C. Sercel [4], Luisa Whittaker-Brooks[2] &
Z. Valy Vardeny [1✉]

The Rashba splitting in hybrid organic–inorganic lead–halide perovskites (HOIP) is particu-
larly promising and yet controversial, due to questions surrounding the presence or absence
of inversion symmetry. Here we utilize two-photon absorption spectroscopy to study
inversion symmetry breaking in different phases of these materials. This is an all-optical
technique to observe and quantify the Rashba effect as it probes the bulk of the materials. In
particular, we measure two-photon excitation spectra of the photoluminescence in 2D, 3D,
and anionic mixed HOIP crystals, and show that an additional band above, but close to the
optical gap is the signature of new two-photon transition channels that originate from the
Rashba splitting. The inversion symmetry breaking is believed to arise from ionic impurities
that induce local electric fields. The observation of the Rashba splitting in the bulk of HOIP
has significant implications for the understanding of their spintronic and optoelectronic
device properties.

[1] Department of Physics & Astronomy, University of Utah, Salt Lake City, UT 84112, USA. [2] Department of Chemistry, University of Utah, Salt Lake City, UT
84112, USA. [3] Sivananthan Laboratories, Bolingbrook, IL 60440, USA. [4] Center for Hybrid Organic Inorganic Semiconductors for Energy, 15013 Denver West
Parkway, Golden, CO 80401, USA. ✉email: val@physics.utah.edu

The splitting of spin angular momentum in electronic systems in the presence of spin-orbit coupling (SOC) and lack of inversion-symmetry is known as the Rashba–Dresselhaus effect. The Dresselhaus effect, as described in zinc blende semiconductors[1], originates from bulk crystal structures that lack inversion symmetry, while the Rashba effect, originally described in wurtzite crystals[2], and reformulated in the Bychkov–Rashba model for 2D electron gases[3], results from inversion asymmetry that arises from site point group dipoles of constituent atoms that add up to a non-zero vector that can point in the direction of either a bulk inversion asymmetry ("bulk Rashba materials") or a structural inversion asymmetry such as in the case of surfaces or hetero-junctions in quantum wells[4]. The possibility of coupling between electric fields and spin degrees of freedom that can result from this effect forms the basis for a variety of spintronics applications[5–7] including the Datta–Das spin transistor[8], the spin Hall effect[9], and the Edelstein effect[10]. Therefore, a great deal of recent interest has been given to the development and characterization of materials that show Rashba splitting, with much progress made in systems such as semiconductor heterojunctions[11] and quantum wells[12], surfaces of metals, alloys, and oxides[13–15], and more recently in 3D bulk ionic semiconductors[16–18]. In contrast, the breadth of experimental techniques available to observe and characterize the spin splitting has been severely narrow during this development period.

The hybrid organic–inorganic perovskites (HOIP) are one of the most promising material families that may harness electrical or optical control over spins in solid-state materials. Previous theoretical works discussed the possibility of the Rashba effect in the MAPbX$_3$ perovskites, where MA is methyl ammonium and X is a halide atom such as Bromine (Br) or Iodine (I)[19,20]. The presence of Pb atoms imparts a large spin–orbit interaction in the electronic structure. On the other hand, the mechanism of inversion symmetry breaking in these materials is less clear. As the crystal structure exhibits phase transitions as temperature (or pressure) is varied from the high-temperature cubic phase, to tetragonal and orthorhombic phases at lower temperatures[21,22], it has been suggested that certain ferroelectric phases may display Rashba splitting[19,20]. Others have considered the motional free-dom of the A-site cations such as MA or Cs that may result in local breaking of inversion symmetry[23,24]. Direct experimental confirmation of Rashba splitting from the surface of MAPbBr$_3$ crystals was observed by ARPES measurements[25], though in this case, the surface termination itself may be the root of the broken symmetry. On the other hand, the second harmonic generation polarization analysis revealed an inversion symmetric phase for MAPbI$_3$ at room-temperature (RT) and hence the absence of Rashba splitting[26].

Here we use two-photon absorption (TPA) spectroscopy to clarify the role of inversion symmetry breaking and its relation to the Rashba effect in the bulk of HOIP. We show that inversion symmetry breaking and Rashba-splitting are correlated with the presence of ionic defects in the crystals and can thus be attributed to local electric fields. Specifically, we demonstrate Rashba split-ting in single crystals of MAPbI$_3$ at ambient and cryogenic temperature, and show that Rashba splitting is induced in MAPbBr$_3$ during the transition to the low-temperature orthor-hombic phase. Additionally, we observe the signature of the Rashba effect in single crystals of MAPbI$_x$Br$_{3-x}$ and in two-dimensional (2D) perovskite phenyl ethyl ammonium lead iodide (PEA$_2$PbI$_4$) due to intrinsic inversion symmetry breaking that results from halide alloying and 2D layered structure, respectively.

For our measurements, we have used two-photon photo-luminescence excitation spectra (TP-PLE) in a broad spectral range from 0.8 to 1.5 eV. We first explain how TPA provides a unique tool for the evaluation of the presence or lack of inversion

symmetry in semiconductors, as well as the observation of Rashba energetic splitting. The former is manifested in the forbidden or allowed nature of TPA in the exciton spectral region, whereas the latter arises due to extra intra-band transition channels that become available for materials with Rashba splitting that results in a strong TPA band at energies determined by the strength of the splitting. We then show experimental spectra in the HOIP specified above, thereby including systems in which Rashba splitting has previously been observed or excluded. Furthermore, the large penetration depth of the below-gap photons used here ensures that the bulk properties are examined, making this technique particularly suitable for investigating bulk electronic materials. Our present observation of Rashba splitting, therefore, demonstrates the potential for HOIP to be designed to function as 'bulk' Rashba materials similar to the family of BiTeX materials[16–18]. The confirmation of Rashba splitting may also have profound implications in efforts to develop the lead–halide perovskites in optoelectronic devices such as LEDs and solar cells[23,24,27].

## Results and discussion

TPA is a nonlinear optical phenomenon described by the ima-ginary part of the third-order susceptibility $\beta(\omega) \propto \text{Im}[\chi^{(3)}]$ that may be considered as an instantaneous sequential process, wherein the first photon excites an electron to a virtual inter-mediate energy level, while the second photon excites the same electron to the real final energy level[28]. As such, this process is described by 2nd order perturbation theory, whereby the first electric dipole operator couples the initial occupied state to an unoccupied intermediate state, and the second dipole operator couples an initially unoccupied intermediate state to an initially unoccupied final state. This has significant consequences regarding the allowed transitions for both exciton and band states that depend crucially on the symmetry of the crystal, and in particular whether parity is a good quantum number.

For systems with inversion symmetric crystal structures, parity is a good quantum number leading to a distinct partitioning between the one and two-photon allowed states. This is the case because the electric dipole operator can only couple the symmetry crystal ground state to exciton states with odd total parity. For example, for a cubic crystal with O$_h$ symmetry the ground state with $\Gamma_1^+$ symmetry (in the nomenclature of Koster, Dimmock, Wheeler, & Statz[29]), has 'even' parity and only an 'odd' parity exciton with $\Gamma_4^-$ symmetry can serve as an intermediate exciton state. From there, only an 'even' parity exciton can serve as a final state. Therefore, the final state reached by a two-photon transition is one-photon forbidden and likewise, the lowest energetic exci-ton state is one-photon allowed and two-photon forbidden. In a monoclinic crystal with inversion symmetric C$_{2h}$ point group a similar situation exists, where only the odd-parity, one-photon allowed exciton states may serve as intermediates to the even-parity, two-photon allowed final exciton states (see Supplemen-tary Note 1 and Supplementary Figs. 1 and 2). Such a distinct exciton partitioning allows TPA to be a very useful probe of exciton properties and was used in this regard to provide direct proof of the strong role of electron correlation effects in carbon nanotubes, transition metal dichalcogenides, and π-conjugated polymers[30–32]. When inversion symmetry is broken, however, parity no longer serves as a good quantum number and such strict partitioning may not occur. For example, for the two sys-tems discussed, the loss of inversion symmetry along a single direction in a cubic system leads to C$_{4v}$ point symmetry where a two-photon allowed exciton state with $\Gamma_1$ symmetry is one-photon allowed when polarized along the direction of symmetry breaking. More drastically, in a monoclinic system that belongs to the C$_s$ point group that lacks inversion symmetry, all exciton

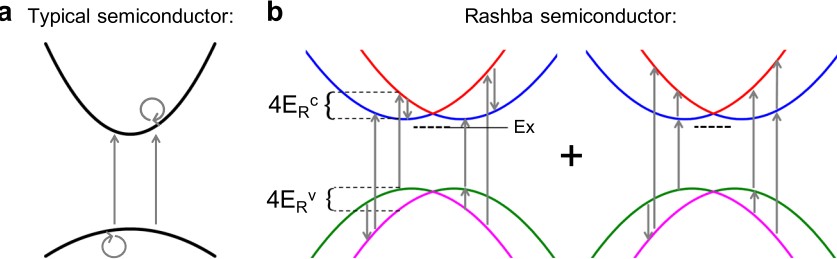

**Fig. 1 Schematic illustration of the two-photon pathways for semiconductors with and without Rashba splitting. a** Two-photon pathways for transitions between initial, intermediate, and final states limited to highest valence and lowest conduction bands in a typical semiconductor. **b** Examples of additional two-photon pathways for transitions between initial, intermediate, and final states available in a semiconductor that exhibits Rashba splitting. The energy of the Rashba splitting in the conduction band ($E_R{}^c$) and valence band ($E_R{}^v$), as well as the exciton (Ex), are shown.

states are both one and two-photon allowed (see Supplementary Note 1 and Supplementary Figs. 3 and 4). Thus, in a general sense we may monitor the presence or absence of symmetry breaking in a given semiconductor by the presence or absence of an overlap between one- and two-photon allowed transitions at $\hbar\omega$ and $2\hbar\omega$, respectively. For the band transitions, similar parity selection rules hold, although only at high-symmetry points in the Brillouin zone (BZ)[33].

We next consider how TPA additionally allows us to observe Rashba-type band dispersion and determine the associated energetic splitting. Figure 1a depicts a schematic illustration of the transition pathways associated with TPA. Intermediate transitions are not subject to conservation of energy and therefore may involve higher lying conduction bands. However, it was shown by Van Stryland and coworkers that only the lowest valence band (VB) to conduction band (CB) transitions were necessary to describe TPA across a wide range of semiconductors[34]. Because the photon momentum is negligible on the scale of the BZ, both the intermediate and final transitions conserve crystal momentum **k** and are represented by vertical lines. The schematic in Fig. 1a reveals that for resonant TPA processes all final states reside in the CB and only intra-band transitions may contribute as intermediates (see Supplementary Note 2 and Supplementary Fig. 5). This condition produces the typical spectral response of the TPA process in semiconductors[34]. If, on the other hand, the bands are split by the Rashba effect, two significant deviations from the typical situation occur, as illustrated schematically in Fig. 1b. First, there are now two energetically distinct bands of final states. In addition, the number of available pathways for transitions increases leading to an overall enhancement of the TPA process. Specifically, the lower (upper) branches of the split CB may serve as intermediate states for two-photon transitions to the upper (lower) branches. This is in addition to the more typical process where intra-band transitions serve as intermediate states (see Supplementary Fig. 6). We further point out that optical transitions between Rashba split branches have been observed in both BiTeX and 2D HOIP systems and are spin-allowed[35,36]. Hence, the resulting spectrum will show a double-peaked structure, with maximal TPA occurring when $2\hbar\omega$ is above the band-gap energy $E_G$ by an amount determined by $E_R{}^{c(v)}$, the Rashba energy splitting associated with the conduction (valence) bands (see Supplementary Note 3 and Supplementary Figs. 7–9). The doubly-peaked TPA spectrum, in conjunction with the observation of inversion symmetry breaking by the two-photon allowed exciton spectrum, thus provides a signature of Rashba splitting in the bulk of the material. Meanwhile, the energetic separation of the peak in TPA from the band-gap energy allows the determination of the Rashba splitting energy. In what follows, we demonstrate these effects experimentally in representative examples from the HOIP family.

The most direct approaches for measuring TPA are the z-scan technique[37], or via intensity-dependent nonlinear-transmission[34]. However, for thick samples such as perovskite single crystals, scattering of the optical transmission makes reliable measurements difficult, and obtaining spectral information can be extremely time consuming using these techniques. Here we use the fact that following multiphoton excitation processes such as TPA, electrons and holes may recombine to produce two-photon photoluminescence (TP-PL). We thus monitor the TP-PL as a function of the incident photon energy, $\hbar\omega$ to obtain the TP-PLE spectra. Figure 2 shows our experimental setup. We used the tunable wavelength output of an optical parametric oscillator (OPO) seeded by a Q-switched Nd:YAG laser that delivers 7 ns pulses at 10 Hz repetition rate for exciting crystals of HOIP (see "Methods" for crystal growth and experimental details). We show in Fig. 2a the resulting PL spectra of a MAPbBr$_3$ crystal following one-photon and two-photon excitation, respectively. The TP-PL band is considerably red-shifted with respect to that of the one-photon PL. Previous reports have attributed this to strong reabsorption in the bulk of the crystal from which it originates[38,39], as the penetration depth for excitation below the bandgap, $E_G$ is much greater than that above $E_G$, where strong linear absorption is present. For TPA, the penetration depth is determined by the inverse of the product of the pump intensity and TPA coefficient $\beta$. In our case, the penetration depth is of the order of 1 mm. A similar difference between the PL and TP-PL spectra of MAPbI$_3$ and PEA$_2$PbI$_4$ is shown in the Supplementary Information (see Supplementary Note 4 and Supplementary Fig. 10). We further note that the one-photon PLE spectra of these compounds are featureless, exhibiting only a decrease in PL with increasing photon energy due to enhanced nonradiative surface recombination (see Supplementary Note 5 and Supplementary Fig. 11). That the TP-PL originates from two-photon excitation is confirmed by the quadratic dependence on the pump intensity, as shown in Fig. 2b. At excitation photon energy, $\hbar\omega = 2.80$ eV, which is above the bandgap, one-photon absorption occurs. Consequently, from the slope of the intensity dependence on a log–log scale, we obtain a pump dependence exponent, $\gamma$, as defined by the relation PL $\propto I_p^\gamma$ of 0.6; beneath the expected linear relationship, probably due to bimolecular recombination which is stronger at high excitation concentration. We further obtain $\gamma = 2.0$ at $\hbar\omega = 1.24$ eV, and only a slightly lower $\gamma$ of 1.9 at $\hbar\omega = 1.50$ eV. These values indicate a consistent TPA process throughout this energy region.

Integrating the PL spectrum and plotting the result vs. $2\hbar\omega$, we obtained the TP-PLE spectrum as shown for the MAPbBr$_3$ crystal at room temperature in Fig. 3a. We compare the TP-PLE spectrum against the absorption spectrum measured from a MAPbBr$_3$ thin film. The relationship between $E_G$ of thin films and bulk crystals is discussed in Supplementary Note 4. We note that the

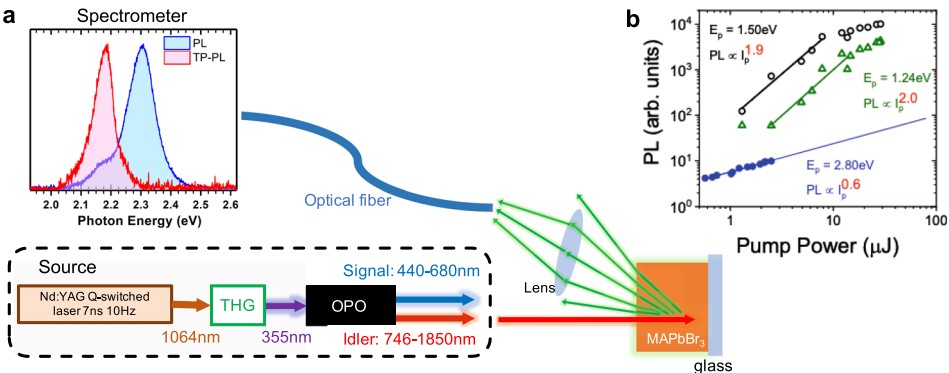

**Fig. 2 Measurement of two-photon photoluminescence of hybrid organic–inorganic perovskite crystals. a** Schematic of the experimental set-up in which the spectrally tunable output of an optical parametric oscillator (OPO) seeded by the combination of an Nd:YAG Q-switched laser and third-harmonic generator (THG) is used as excitation light source for the photoluminescence (PL) emission from the perovskite crystals. The PL emission spectrum from MAPbBr$_3$ crystal is shown for one-photon excitation (blue) and two-photon excitation (red). **b** Log–log plot of the pump intensity ($I_p$) dependence of the PL emission at excitation photon energies ($E_p$) of 1.24 eV (green open triangles), 1.50 eV (black open circles), and 2.80 eV (blue solid circles). Solid lines are linear fits to extract the power law as indicated.

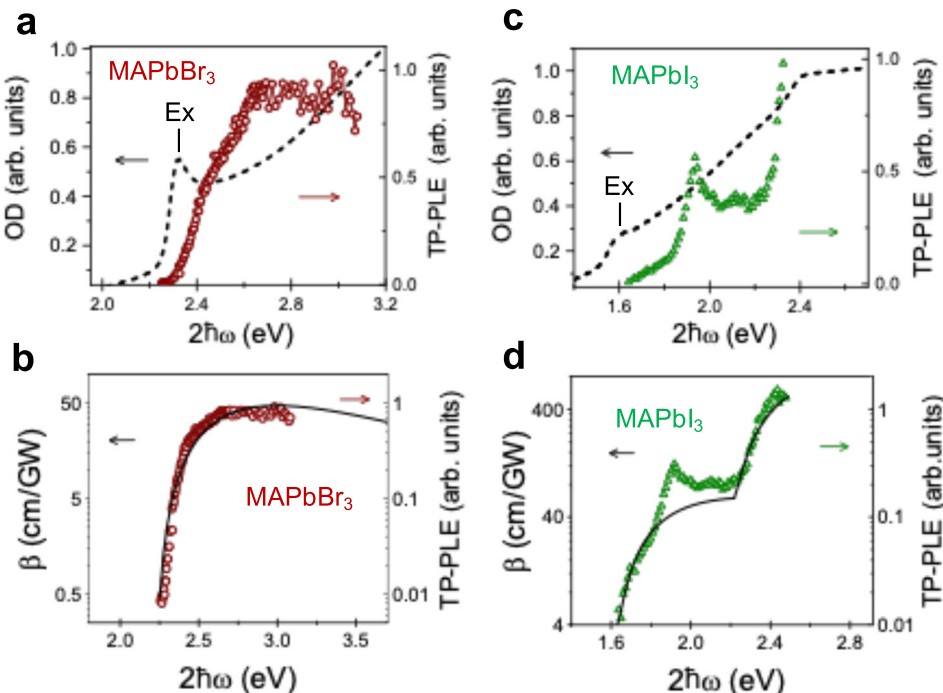

**Fig. 3 Two-photon photoluminescence excitation (TP-PLE) spectra of HOIP crystals measured at room temperature compared to the absorption spectra and model calculations.** TP-PLE spectrum of MAPbBr$_3$ (brown circles) compared to **a** the absorption spectrum of the corresponding thin film (dashed line) and **b** theoretical calculations for the TPA response $\beta(\omega)$ excluding the Rashba effect (solid black line) plotted on a semi-log scale. TP-PLE spectrum of MAPbI$_3$ (green triangles) compared to **c** the absorption spectrum of a thin film (dashed line) and **d** theoretical calculations for the TPA response $\beta(\omega)$ excluding the Rashba effect (solid black line) plotted on a semi-log scale. The TP-PLE spectra are plotted vs. twice the excitation energy, $2\hbar\omega$; whereas the absorption spectra are plotted vs. $\hbar\omega$. The exciton region (Ex) is indicated in (**a** and **c**).

TP-PLE is plotted vs. $2\hbar\omega$ whereas the absorption is plotted vs. $\hbar\omega$, so that the total transition energy involved in the two processes may be compared. It can be seen that the TP-PLE spectra at $2\hbar\omega$ of MAPbBr$_3$ crystals are very weak in the vicinity of the absorption onset (near 2.3 eV); instead, it shows a stronger onset at $2\hbar\omega = 2.4$eV and steadily increases at larger energies reaching a plateau at $2\hbar\omega = 2.7$ eV.

In Fig. 3b we compare the measured TP-PLE spectrum in MAPbBr$_3$ to the calculated $\beta(\omega)$ response using standard 2nd order perturbation theory without including the Rashba effect (solid lines in Fig. 3b). The initial state is the ground state with energy $E_i$ plus two photons and the final state is a real excited

state of the material with energy $E_f$. Conservation of energy requires $E_f = E_i + 2\hbar\omega$, with $\hbar\omega$ the excitation photon energy. The intermediate state is a virtual state with energy $E_t$. The probability of absorbing two photons is

$$W = \frac{2\pi}{\hbar}\left(\frac{4\pi^2 e^4 I_p^2}{n_p^2 c^2 m^4 \omega^4}\right) \int \sum_f \left|\sum_t \frac{M_{ft}M_{ti}}{E_f - E_t - \hbar\omega}\right|^2 \delta\left(E_f - E_i - 2\hbar\omega\right)\frac{d^3k}{(2\pi)^3}$$

(1)

where $I_p$ is the pump intensity, $n_p$ is the refractive index at the pump photon energy, $m$ is the free electron mass, and $M_{ft}$ and

$M_{ti}$ are the transition matrix elements

$$M_{ft} = \langle f | \boldsymbol{\varepsilon} \cdot \boldsymbol{p} | t \rangle, \quad M_{ti} = \langle t | \boldsymbol{\varepsilon} \cdot \boldsymbol{p} | i \rangle, \qquad (2)$$

with $\boldsymbol{\varepsilon}$ the polarization vector of the pump beam, and $\boldsymbol{p}$ the momentum operator. The TPA coefficient $\beta(\omega)$ is defined as $\beta(\omega) = 2\hbar\omega W / I_p^2$ in units of cm/GW. Exciton effects were included using a Green's function approach assuming a hydrogenic spectrum (see Supplementary Note 6). The line shape is consistent with the typical TPA response of semiconductors[34]. Here, satisfactory agreement between the calculated spectrum and the experimental data can be obtained using a VB→CB interband transition including excitonic enhancement effects (see Supplementary Fig. 12).

In Fig. 3c we show the MAPbI₃ TP-PLE spectrum compared to the absorption of a thin film. Similar to the case of MAPbBr₃, we find that the TP-PLE here is also very weak at $2\hbar\omega \gtrsim E_g$. However, there are considerable differences between the TP-PLE spectrum of MAPbI₃ and MAPbBr₃. Most notably, the former contains a sharp band peaking at $2\hbar\omega = 1.90$ eV that has no corresponding feature in linear absorption. We attribute this peak to transitions between Rashba-split conduction and valence bands as discussed below. As the pump excitation energy $\hbar\omega$ increases, a second band is clearly resolved above $2\hbar\omega \approx 2.20$ eV. A corresponding feature in the absorption spectrum is observed at approximately at the same energy. Such a second band has been observed by numerous methods in the literature, including absorption[40], ellipsometry[41], and pump-probe spectroscopy[42], and most interpretations have considered a transition involving higher energy conduction-bands, or a contribution from a valley along another symmetry axis in the BZ[41,43].

In contrast to MAPbBr₃, the lower energy band in the measured TP-PLE spectrum of MAbI₃ is well accounted for by traditional band-to-band transition only at low energy and deviates significantly above $2\hbar\omega = 1.80$ eV because of the additional absorption as clearly demonstrated in Fig. 3d. including relatively sharp features that become much clearer at low temperature as shown below. Additionally, to model the TP-PLE spectrum in MAPbI₃ requires the inclusion of an additional, higher energy inter-band transition at $2\hbar\omega = 2.20$ eV, consistent with the energy difference between the VB and the higher lying CB (CB2) (split-off band). Wei et al. also observed this higher energy band using two-photon induced microwave conductivity and similarly attributed it to the contribution from the split-off conduction band[44]. This second band at higher energy is also well described by standard 2nd order perturbation calculations only at energies near this band onset. We note that for MAPbBr₃, the measured TP-PLE response has a contribution only from one band (CB1), as the second transition VB → CB2 occurs outside of the TPA experimental spectral range, as verified by Electroabsorption measurements (see Supplementary Note 7 and Supplementary Fig. 13).

To correctly understand the deviations of the TP-PLE spectrum of MAPbI₃ from the standard model requires the inclusion of the Rashba effect. Incorporating this effect, the Hamiltonians for the conduction and valence bands can be written,

$$H_{c\boldsymbol{k}} = \frac{\hbar^2}{2m_e}\left(k_x^2 + k_y^2 + k_z^2\right) + \alpha_c\left(k_y\sigma_x - k_x\sigma_y\right) \qquad (3)$$

$$H_{v\boldsymbol{k}} = -E_G - \frac{\hbar^2}{2m_h}\left(k_x^2 + k_y^2 + k_z^2\right) + \alpha_v\left(k_y\sigma_x - k_x\sigma_y\right) \qquad (4)$$

The Rashba effect lifts the spin degeneracy and splits the lowest conduction band ($c$) into upper and lower bands ($c\pm$) and the valence band ($v$) is split into upper and lower bands ($v\pm$). In the presence of the Rashba effect, the velocity operator for the conduction and valence bands must be reevaluated from the Hamiltonians $H_{c\boldsymbol{k}}$ and $H_{v\boldsymbol{k}}$:

$$\boldsymbol{v}_{c(v)} = \dot{\boldsymbol{r}} = \frac{1}{i\hbar}\left[\boldsymbol{r}, H_{c(v)}\right] = \frac{\hbar}{2\sqrt{m_{e(h)}}}\left(e_+k_- + e_-k_+\right) + i\sqrt{2}\frac{\hbar k_{0c(v)}}{m_{e(h)}}\left(e_-\sigma_+ - e_+\sigma_-\right)$$

$$(5)$$

where $e_\pm = \frac{1}{\sqrt{2}}\left(e_x \pm ie_y\right)$ and $k_\pm = k_x \pm ik_y$ and $\sigma_\pm = \sigma_x \pm i\sigma_y$. The first two terms are the common velocity, $\boldsymbol{v}_{c(v)} = \hbar\boldsymbol{k}/m_{e(h)}$ in a parabolic band, which preserves (pseudo)spin. The second terms, however, are induced by the Rashba effect.

Labeling the bands in order of increasing energy as 1–4, i.e., $v_- = 1$, $v_+ = 2$, $c_- = 3$, and $c_+ = 4$, (Supplementary Fig. 8) and denoting the four final states by the location of the resulting electron and hole, (e.g., (3, 2) for the final state with electron in band 3, and hole in band 2), we have the four possibilities (3, 1), (3, 2), (4, 1) and (4, 2). The transition rate for a given state, say (4,1) is then

$$W_{(4,1)} \propto \left|\sum_{t=1}^4 \frac{\langle 4 | \boldsymbol{\varepsilon} \cdot \boldsymbol{p} | t \rangle \langle t | \boldsymbol{\varepsilon} \cdot \boldsymbol{p} | 1 \rangle}{E_t - E_1 - \hbar\omega}\right|^2 \delta\left(E_4 - E_1 - 2\hbar\omega\right) \qquad (6)$$

In each of the four terms, one of the two matrix elements represents an *inter*-band transition while the other represents an *intra*-band transition. The inter-band terms $p_{cv}$ are large and independent of the Rashba effect. Whereas the *intra*-band terms depend on the Rashba effect which can be evaluated using the velocity operators. In the absence of the Rashba effect the matrix elements $\langle 4 | P_v | 3 \rangle$ and $\langle 2 | P_v | 1 \rangle$, where $P_v = -[e/mc]\boldsymbol{A} \cdot \boldsymbol{p} = -[e/c]\boldsymbol{A}\boldsymbol{v}$ describes the electron-photon interaction, are null due to the opposite spins in the conduction and valence sub-bands. In the presence of the Rashba effect that mixes spin and orbital momentum, these additional "spin-flip" terms become available.

First, we demonstrate the experimental observation of the Rashba effect by considering the TP-PLE spectrum of MAPbBr₃ measured at a temperature of 50 K. As seen in Fig. 4a the TP-PLE spectrum contains an additional band on top of the usual inter-band contribution, which shows the characteristic Rashba signature and enhanced TPA at this temperature. We estimate the Rashba strength from the comparison of the experimental data to model calculations based on the Hamiltonian including the Rashba term as shown in Fig. 4a (see also Supplementary Note 3 and Supplementary Fig. 9). As seen, the model exhibits satisfactory agreement when $\alpha_R = 3$ eVÅ. We further note that the low-energy shoulder is a signature of the inversion symmetry breaking and subsequent two-photon optically allowed exciton transitions. A similar behavior was observed in TPA spectra of GaAs[45]. In fact, the TP-PL band of this crystal shows that $E_g$ occurs at higher energy at 50 K than at room temperature (see Supplementary Note 8 and Supplementary Fig. 14a), as does the value of $E_G$ extracted from the comparison of the TP-PLE spectra to the standard model for TPA without Rashba splitting, so the lower-energy onset of the TP-PLE spectrum at 50 K shows substantially stronger allowed TPA at $2\hbar\omega \approx E_G$ than at RT as a consequence of the symmetry breaking.

We also performed intensity-dependent transmission measurements to extract a quantitative TPA value for this crystal at both RT and at 50 K (Fig. 4b, c), which allows us to properly compare the magnitude as well as the spectral line shape of the TPA. As seen in Fig. 4a, there is indeed an increase in the obtained $\beta$ at 50 K due to the Rashba effect that is predicted by the model. The measured values at room temperature are consistent with those previously reported[46–48]. Importantly, the difference between $E_g$ at the two temperatures cannot account for the increase in $\beta$ through the traditional band-gap scaling law for TPA, which states that the magnitude of $\beta$ scales as $E_G^{-3}$[34,49]; because this trend would suggest that $\beta$ is smaller at this temperature, contrary to what is observed.

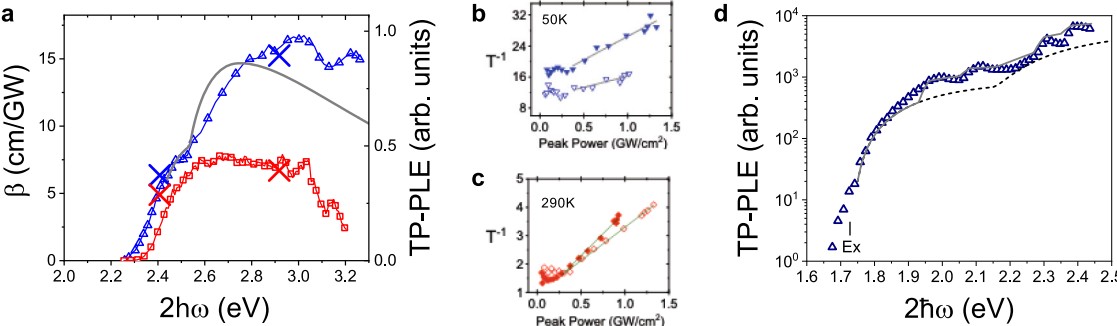

**Fig. 4 Rashba splitting in MAPbBr3 and MAPbI3 crystals at low-temperature observed by two-photon photoluminescence excitation (TP-PLE) spectroscopy. a** TP-PLE spectrum of MAPbBr$_3$ at a temperature of 50 K (blue triangles) compared to the room temperature (290 K) spectrum (red squares). The gray solid line is $\beta(\omega)$ calculated including the Rashba effect. The X's on the experimental data indicate the measured values of $\beta$ using the intensity-dependent transmission measurements shown in (**b** and **c**). **b, c** Inverse transmission ($T^{-1}$) vs. peak pulse power at $2\hbar\omega = 2.4$ eV (open symbols) and $2\hbar\omega = 2.9$ eV (closed symbols) at temperature of 50 K (**b**) and 290 K (**c**). The solid lines are linear fits to the data with slopes proportional to $\beta$. **d** TP-PLE spectrum of MAPbI$_3$ at a temperature of 77 K (blue triangles) plotted on a semi-log scale. The lines are $\beta(\omega)$ calculated excluding (dashed black) and including (solid gray) the Rashba effect. The position of the exciton (Ex) is indicated.

Other possibilities for the occurrence of an additional TPA band at this photon energy can be ruled out. Firstly, the higher-lying conduction bands occur at much higher energy. Electro-absorption measurements at RT place the 2nd conduction band (CB2) of MAPbBr$_3$ at or above 3.3 eV. Meanwhile, the energetic position of CB2 does not change much with temperature since it is determined by the SOC and is therefore weakly dependent on the crystal phase[50]. Furthermore, the presence of trap states or polaronic absorption would give rise to features below the bandgap that are not experimentally observed, and thus cannot explain the additional feature above the bandgap. Meanwhile, trap-state enhanced TPA would require states that lie near but slightly above the mid-gap, while theoretical predictions place such levels very near the conduction band minimum[51].

In Fig. 4d, we show the TP-PLE spectrum of MAPbI$_3$ at 77 K and compare it to the model including the Rashba effect. The spectrum at 77 K is similar to that at RT, but due to the higher PL efficiency at low temperature the features are much more clearly resolved. We observe a slight shoulder near $2\hbar\omega = 1.65$ eV that is attributed to the partially allowed exciton transition due to symmetry breaking. The transitions into CB1 are then observed at higher energy following the standard model until $2\hbar\omega = 1.80$ eV. Above this energy, TPA increases much faster than predicted by the standard model but is captured by modeling that includes the Rashba effect. Here we clearly resolve two peaks at $2\hbar\omega = 1.94$ eV and $2\hbar\omega = 2.06$ eV. Fitting these peaks to the model described above (see also Supplementary Note 3 and Supplementary Fig. 9) give values of the respective Rashba energies of $E_R^v \approx 0.008$ eV for the VB and $E_R^{c1} \approx 0.332$ eV, for CB1. Also similar to the spectrum at RT, we observe at 77 K a second onset of TPA attributed to transitions to CB2 that occurs at $2\hbar\omega = 2.15$ eV, which again is only partly described by the standard model, and also contains two additional peaks at $2\hbar\omega = 2.28$ eV and $2\hbar\omega = 2.37$ eV, respectively. The model fitting gives $E_R^{c2} \approx 0.214$ eV for the energetic splitting in the second conduction band. We note that the Rashba splitting in higher continuum bands has never been experimentally verified before, demonstrating the power of the TP-PLE spectroscopic technique.

As the bulk Rashba effect must arise from a net sum of site asymmetries (that is local inversion symmetry breaking)[4], the controversy over the presence of Rashba splitting in perovskites has focused on the source of inversion symmetry breaking. The general consensus is that these materials crystallize in inversion symmetric phases at all temperatures. In our case, we have

observed a correlation between the observation of the signature of Rashba splitting in the TP-PLE spectrum and a spectrally dependent TP-PL pump intensity dependence. Namely, the intensity dependence exponent, $\gamma$ is found to increase to values larger than 2 as $2\hbar\omega$ decreases towards $E_G$ (see Supplementary Note 9 and Supplementary Fig. 15). The super-linear photoluminescence intensity dependence has been observed in one-photon photoluminescence measurements of HOIP[52] and is due to exciton recombination that occurs in the presence of donor or acceptor states, and is thus evidence of ionic defects and impurity states[53]. Moreover, we found that in MAPbBr$_3$ at room temperature the power law coefficient $\gamma = 2$ and is constant across the whole spectrum. This indicates that the exciton recombination is monomolecular, and does not proceed via charge defects. This should be correlated with a lack of inversion symmetry breaking and in turn Rashba splitting at room temperature. We consequently attribute the presence of inversion symmetry breaking in three-dimensional HOIP to these ionic defect states. In the case of MAPbBr$_3$, these defects may be induced as the crystal goes through the temperature-induced phase transitions, whereas in MAPbI$_3$ they are present even in the room temperature phase leading to little change as temperature is lowered.

In order to further test this hypothesis, we examined the TP-PLE spectrum of mixed-halide perovskite crystals, MAPbBr$_{3-x}$I$_x$ with low-iodide content, $x$. In doing so, we avoid inducing a phase transition from the cubic to tetragonal phase, while explicitly breaking inversion symmetry through the mixing of halide anions of different size. The results are shown in Fig. 5a. In comparison to room temperature MAPbBr$_3$, the TP-PLE spectra obtained from MAPbBr$_{2.7}$I$_{0.3}$ and MAPbBr$_{2.4}$I$_{0.6}$ show distinct features consistent with the method described herein for using-TPA to observe inversion symmetry breaking and the Rashba effect. Firstly, we observe the shoulders near the onset of TP-PLE at 2.18 eV for MAPbBr$_{2.4}$I$_{0.6}$ and at 2.24 eV MAPbBr$_{2.7}$I$_{0.3}$ that demonstrate the two-photon allowed nature of the exciton transition via inversion symmetry breaking. The identification of these features as exciton peaks is confirmed by their energies which directly coincide with the peak position of the one-photon photoluminescence from these crystals (see Supplementary Note 10 and Supplementary Fig. 16). Secondly, we observe a pronounced feature that occurs above the band-gap consistent with the observation of the transitions between levels of Rashba-split valence and conduction bands. The positions of these bands are found to occur at 0.16 and 0.12 eV above the

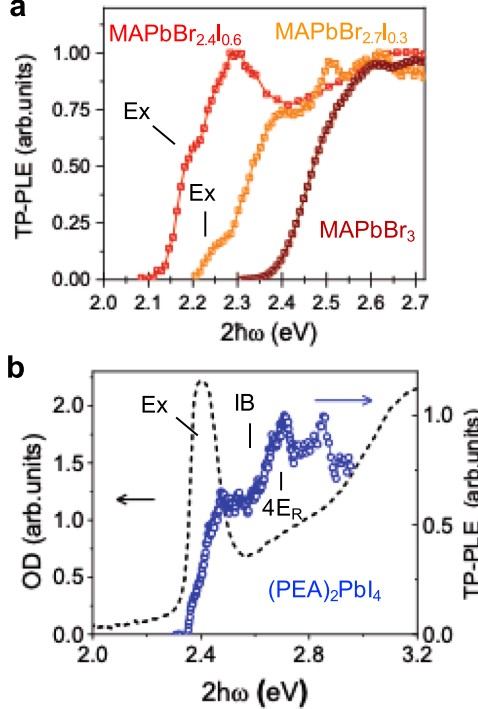

**Fig. 5 Broken inversion symmetry and Rashba splitting in mixed-halide perovskite crystals and 2D PEA₂PbI₄.** **a** Room temperature two-photon photoluminescence excitation (TP-PLE) spectra of mixed halide crystals MAPbBr$_{2.4}$I$_{0.6}$ (red squares) and MAPbBr$_{2.7}$I$_{0.3}$ (orange squares) compared to that of MAPbBr$_3$ (brown squares). **b** TP-PLE spectrum of (PEA)$_2$PbI$_4$ (blue circles) compared to the absorption spectrum of a corresponding thin film (dashed line). The TP-PLE spectra are plotted vs. twice the excitation energy, $2\hbar\omega$, whereas the absorption is plotted vs. $\hbar\omega$. The positions of the exciton (Ex) are indicated. In **b**, the peak due to Rashba splitting ($4E_R$) and the inter-band onset (IB) are also designated.

exciton position for MAPbBr$_{2.7}$I$_{0.3}$ and MAPbBr$_{2.4}$I$_{0.6}$ respectively.

We also demonstrate both inversion symmetry breaking and the Rashba effect in the TP-PLE spectrum of 2D PEA$_2$PbI$_4$ as seen in Fig. 5b. In this case, two-photon excitation is strongly allowed within the energy range of the lowest lying exciton at $2\hbar\omega \approx 2.4$ eV. The alternating layers of organic and inorganic components, as well as octahedral distortions in this material result in inversion symmetry breaking and the optical selection rules for $C_s$ point group apply. Meanwhile, the TP-PLE spectrum increases to a maximum at 2.72 eV. Comparison with previous studies, place this peak at 0.14 eV above the inter-band absorption onset at 2.58 eV[35], which can also be observed as an increase in TP-PLE above the lower energy exciton shoulder, revealing a Rashba energy of $E_R \approx 0.035$ eV, a value that is consistent with independent determination by photo-modulation and circular photogalvanic spectroscopy[35,54].

In conclusion, we report on an all-optical method for observing Rashba band dispersion in semiconductors which is based on TPA spectroscopy. This method is uniquely suited to observe bulk Rashba splitting such as is present in the hybrid perovskites studied here. We observed a correlation between inversion symmetry breaking in 3D perovskites with the presence of defects that may produce local electric fields. These defects are present naturally in our MAPbI$_3$ crystals and can be directly engineered by alloying halide anions in MAPbBr$_{3-x}$I$_x$ crystals. On the other hand, defects are induced upon structural distortion during the

temperature-driven phase transition as in low temperature MAPbBr$_3$. The mechanism we have elucidated occurs in addition to the natural inversion symmetry breaking that occurs in the layered 2D perovskite PEA$_2$PbI$_4$. The energetic splittings are found to be on the order of 10's of meV, similar to the values found in BiTeX polar semiconductors[16–18]. Furthermore, we point out that our hypothesis that the origin of symmetry breaking results from ionic defects may help to explain the inconsistent reports about the presence or absence of Rashba splitting in HOIP, since the concentration of these defects may vary depending on the fabrication procedures or exhibit spatial inhomogeneity. Rashba splitting should have a dramatic influence on many properties of these materials including the optical transitions, the unexpected temperature dependence of the carrier mobility[27] and unusually slow recombination rates that are a major advantage for solar cells[23,24,55]. Our observations show the importance of understanding and controlling defect physics for further development of optoelectronic and spintronic applications based on the hybrid perovskites. Particularly, the ability to controllably dope HOIP with optimal defect concentration may be the key to both achieving long, stable carrier lifetimes in photovoltaic devices and for utilizing the Rashba effect in spintronic devices.

## Methods

**Sample preparation.** *MAPbBr$_3$* crystals were grown using an anti-solvent diffusion method in which a precursor solution that consisted of 0.37 g PbBr$_2$ and 0.125 g MABr (Sigma Aldrich) in 2 mL N,N-dimethylformamide (DMF) was first prepared by stirring at 50 °C for 30 min before use. The solution was then allowed to cool to room temperature, after which it was filtered and transferred into an open-top glass vial. The vial was subsequently placed inside a beaker containing 10 mL of 2-propanol as the anti-solvent. The beaker was capped and kept undisturbed for 48 h. The anti-solvent vapors diffused into the precursor solution driving a precipitation-based crystal growth. Mixed-halide crystals were grown by the same method but with the addition of PbI$_2$ and MAI in the proper weight ratio.

*MAPbI$_3$* crystals were grown using a method known as inverse temperature crystallization[56]. 3 mL of a 1.25 M perovskite precursor solution in γ-butyrolactone (GBA) was filtered using a 0.2 μm PET filter and put into a vial. The vial was submerged in an oil bath at 110 °C; crystals were allowed to grow overnight. The crystals were removed from the growth solution after 12 h in the oil bath, dried, and put into a N$_2$ glovebox until completely dry.

*(PEA)$_2$PbI$_4$* crystals were grown by dissolving 1.12 g of PbO in a mixture of 5.00 mL of a 57% w/w hydroiodic acid and 0.85 mL of 50% aqueous H$_3$PO$_2$. The solution was brought to a boil under magnetic stirring for about 5 min, yielding a bright yellow solution. Then, 0.63 mL of 99% phenethylamine was added dropwise while the solution was boiling. The solution was allowed to cool over the course of 2 h during which crystals precipitated out. The crystals were dried and put into an N$_2$ glovebox until completely dry.

*For thin films*, the same precursor solution was used. Films were formed by spin coating on pre-cleaned (O$_2$ plasma exposed for 15 min.) glass substrates at 2000 rpm. To achieve high optical quality films, 0.2 mL of CHCl$_3$ was dripped onto the film during spin-coating, forcing rapid crystallization and resulting in smooth, uniform films with <100 nm crystalline domains. Films were then annealed at 90 °C for MAPbI$_3$ and MAPbBr$_3$ and 100 °C for (PEA)$_2$PbI$_4$ for 20 min on a hot plate.

**Two-photon photoluminescence excitation.** The excitation light source was the variable output of an OPO seeded by the third-harmonic of an Q-switched Nd:YAG laser that generates pulses of 7 ns at a repetition rate of 10 Hz in the spectral range of 0.8–1.5 eV. For photoluminescence measurements, the crystals were excited at normal incidence, and emission was collected at an oblique angle and recorded using a spectrometer. The PL spectra obtained from each excitation energy *was integrated and normalized by dividing the integrated PL by the square of the laser power and multiplying by $2\hbar\omega$ as appropriate for a two-photon process.*

## Data availability

The data that support the findings of this study are available from the corresponding author upon reasonable request.

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

## Acknowledgements

This work was supported by the DOE, Office of Science, grant DE-SC0014579 (PL, MP-PLE, and absorption spectra). Perovskite single crystal synthesis was supported by the DOE under grant# DE-SC0019041. Z.G.Y. was supported partly by the U.S. Army Research Office under Contract No. W911NF-17-1-0511. P.S. and Z.V.V. acknowledge support from the Center for Hybrid Organic Inorganic Semiconductors for Energy (CHOISE), an Energy Frontier Research Center funded by the Office of Basic Energy Sciences. We would like to thank Sumit Mazumdar for helpful discussions regarding the TPA spectroscopy, Qingji Zeng for assistance with intensity-dependent transmission measurements, Sangita Baniya for the electroabsorption measurements, and Xiaojie Liu, Jingying Wang, and Chuang Zhang for additional crystals of MAPbBr3 and perovskite thin films.

## Author contributions

E.L. performed the nonlinear optical measurements, analyzed the data, and wrote the manuscript. Z.-G.Y. performed calculations of two-photon absorption spectra. E.A. grew perovskite crystals and performed XRD characterization. P.S. calculated allowed and

forbidden exciton transitions. L.W.-B. and Z.V.V. supervised the project, discussed research plans, results and data, discussed, and edited the manuscript.

## Competing interests

The authors declare no competing interests.
