## [Peer Review File · Nature Communications]

REVIEWER COMMENTS

Reviewer #1 (Remarks to the Author):

In this manuscript, the authors synthesized single crystals of MAPbX₃ (X = Br and I) and (PEA)₂PbI₄, and studied the band structure of the halide perovskites by performing two-photon absorption spectroscopy. They reported that two-photon photoluminescence excitation spectra can monitor the inversion symmetry breaking and the Rashba band dispersion and they estimated the Rashba splitting energy. The versatile method to observe and estimate the Rashba splitting proposed in this manuscript is important because this method does not need the far-infrared probe light and/or carrier doping. In addition, the solid method to investigate whether the Rashba band dispersion emerges in halide perovskites is required for future improvement of the performance of halide-perovskite-based devices. However, I have significant doubts on the experimental results and their interpretations and further clarification and discussions are required. Therefore, my opinion is that the manuscript is not significant enough to warrant publication in Nature Communications, as it stands.

I suggest some points for improving manuscript:

1. On page 5, the authors mention that the one- and two-photon activities of the exciton transitions can provide the information about symmetry breaking because their parity selection rules are determined by the crystal symmetry. On the other hand, the authors state that the parity selection rules in the band transitions hold only at high-symmetry points in the Brillouin zone. Because the exciton states consist of the band states with different wave vectors, i.e., the states not at the high-symmetry points, why do the parity selection rules of the exciton transitions simply reflect the crystal symmetry?
2. On page 6, the authors mention that the doubly-peaked TPA spectrum provides a signature of Rashba splitting. However, I think that a double peak spectrum can be also obtained if the higher energy band exists. How do the authors distinguish these?
3. Why are the absolute values of β for MAPbBr₃ at room temperature different between Fig. 3b and Fig. 4a? This is a serious issue regarding the credibility of experimental data and analytical models. The absolute values near the bandgap in Fig. 4a are consistent with the recent report for MAPbBr₃. Therefore, I doubt that the absolute values of analytical models are wrong.

4. In Fig.3 c and d, why does the non-monotonic structure, i.e., peaks around 1.6, 1.85, and 2.05 eV, appear in the TP-PLE data below 2.2 eV, which cannot be seen in the theoretical result? Does it reflect the Rashba splitting?

5. In Fig.4a, I doubt the feasibility of the assignment of the additional band on top of the usual interband contribution to the Rashba band structure. Because the trap state and polaron transitions have similar energy scale to the Rashba splitting [ACS Materials Lett., 2, 20–27 (2020)], do the authors comment on that and confirm that the Rashba splitting is the main origin by, for example, performing the polarization-selective spectroscopy?

6. Based on Fig. S13, the authors claim that the bandgap blueshifts at low temperatures. However, this is not acceptable. It is well known that the bandgap of MAPbBr₃ redshifts at low temperature. Therefore, the author's claim that the enhancement of TPA cannot be explained by the traditional band-gap scaling law is also incorrect. The authors should consider that the TP-PL spectrum for perovskite bulk crystal is strongly affected by the reabsorption effect.

7. According to Fig. S13, the linewidth of TP-PL at 50 K is too large (about 70 meV). This suggests that the free exciton emission was strongly reabsorbed and the observed spectrum was dominated by the bound exciton emission. The author should consider the effect of bound exciton on Rashba splitting.

8. On page 9, the authors mention that MAPbBr₃ at 50 K exhibits the Rashba signature. Is it able to mention whether the main contribution stems from surface or bulk?

9. The experimental conditions described in the main text and the materials & methods are different (the laser pulse width and spectral range).

10. The authors say that the TPA spectra β were obtained by integrating the TP-PL spectra from each excitation energy and normalized by the square of the laser power I_0 . However, the obtained spectra should also multiply the excitation photon energy $\hbar\omega$, because the PL intensity is proportional to the excitation carrier density, which equals to $\beta I_0^{>2} \Delta t / 2 \hbar\omega$ where Δt is the excitation pulse length. In addition, the carrier density profile also depends on the value of βI_0 (see, for example, [phys. stat. sol. (b) 245, 2676–2679 (2008)]). The equation in this reference also suggests that if the value of βI_0 is too large, the spectral shape of TP-PL should be changed because of the modification of reabsorption effect. Therefore, the authors should check whether the spectral shapes of TP-PL for each excitation energies were unchanged or not.

11. Is it possible to discuss the contributions of biexcitons or defect states to the observed data?

Reviewer #2 (Remarks to the Author):

In the paper entitled "Broken Inversion Symmetry and Rashba Splitting in Organic-Inorganic Lead-Halide Perovskites Revealed through Two-Photon Absorption Spectroscopy" by Lafalce et al., the authors report an all-optical method for observing Rashba band splitting in several lead-based hybrid perovskites based on two-photon absorption. Using the technique, they claim to observe signatures of bulk Rashba band splitting in methylammonium lead bromide at low temperatures and phenyl ethyl ammonium lead iodide at room temperature. Furthermore, they observe an absence of the Rashba effect in methylammonium lead bromide and methylammonium lead iodide at room temperature.

The hybrid lead-halide perovskites have attracted much attention in recent years for their superior photovoltaic properties. It was initially suggested that broken inversion symmetry (through the associated bulk Rashba effect) could explain the long carrier lifetimes of this class of materials. Since that conjecture, the question of whether these materials maintain or break inversion symmetry has been a controversial issue. I believe this paper could be of great interest to the community by shedding some much-needed light on the problem. I largely find the technique, data, and analysis presented in the paper to be convincing, but I have a few questions that should be addressed by the authors:

(1) Is Rashba splitting observed in methylammonium lead iodide at low temperatures? If the results of this work are to be accepted, an inversion-symmetry-breaking phase transition occurs in the bromide material. It would be interesting to determine if that is a general phenomenon of the hybrid perovskites, or peculiar to this particular member of the family.

(2) The authors claim that their technique is bulk-sensitive. This is a critical aspect of the experiment because the Rashba effect will always exist at surfaces. What is the actual penetration depth of the photons used here, and how has it been determined? Is there any other evidence for the distinction of bulk from surface effects with this technique?

(3) What is the explanation for the apparent change in the power law exponent between low and high power in the green and black curves in Fig. 2b? More specifically, the solid curves are not

straight lines (as claimed in the figure caption), and so do not actually represent any power law. I think it would be less confusing if only straight line fits (i.e. true power laws) were plotted, even if they only match the low power data points.

(4) Line 170: Should be "Fig. 2b", not "Fig. 1b".

(5) The authors should comment on whether their results address the static Rashba effect or the dynamic Rashba effect (i.e. what time scales is this technique sensitive to?).

The following suggestions would help place this work in broader context:

(6) The authors should address if and how their work can potentially explain the sometimes contradictory/inconsistent results in the literature. For example, Ref. [25] claims inversion asymmetry in the bromide material, while Ref. [26] claims the opposite in the iodide.

(7) The authors should emphasize the (important!) consequences of their result. If there is no room-temperature Rashba effect, as they claim in methylammonium lead bromide and iodide, then the theory that it could give rise to the enhanced carrier lifetimes observed in these materials [see for example Refs. [23], [24], or APL Mater. 4, 091501 (2016)] is falsified, and another explanation is needed.

Reviewer #3 (Remarks to the Author):

The manuscript describes two-photon absorption measurements on three halide perovskites MAPbBr₃, MAPbI₃ and the layered materials (PEA)₂PbI₄ presented as evidence for Rashba splitting in those materials. The amount of experimental and theoretical work presented is impressive and extremely interesting, I have, however few comments and questions for the authors before considering publication in Nature Communications:

- I had a doubt reading the manuscript: According to the authors is there or is there not a Rashba signature for MAPbBr₃ at room-temperature (Fig. 3)? It seems they are defending an absence of splitting for MAPbBr₃, but not for MAPbI₃. Is that correct? There should be a very clear sentence in the manuscript to conclude this part.

- The interpretation of the spectra as being evidence of Rashba coupling, relies on the fitting of the experimental data by a model. Sadly, this model is entirely placed in the Supplementary Information document, making the main text not readable by itself. The model should absolutely be discussed in the main text.

- The general consensus is that the low-temperature phase for MAPbX₃ belong to the Pnma group, i.e. a group that presents symmetry of inversion, hence no possible Rashba coupling. However, you find close to record values for the coupling in those materials. This should be discussed in details! Is the Pnma group not a correct choice for those materials? Is there another source of symmetry breaking?

- The shape of the MAPbI₃ spectra has been observed and interpreted differently earlier (Wei et al. Nat. Commun. 10 5342) without invoking Rashba splitting. This should be discussed extensively as it is unexpected for Pnma materials to show Rashba signatures. You need to make sure no other interpretation of the spectra can be made.

REVIEWER COMMENTS

Reviewer #1 (Remarks to the Author):

In this manuscript, the authors synthesized single crystals of MAPbX₃ (X = Br and I) and (PEA)₂PbI₄, and studied the band structure of the halide perovskites by performing two-photon absorption spectroscopy. They reported that two-photon photoluminescence excitation spectra can monitor the inversion symmetry breaking and the Rashba band dispersion and they estimated the Rashba splitting energy. The versatile method to observe and estimate the Rashba splitting proposed in this manuscript is important because this method does not need the far-infrared probe light and/or carrier doping. In addition, the solid method to investigate whether the Rashba band dispersion emerges in halide perovskites is required for future improvement of the performance of halide-perovskite-based devices. However, I have significant doubts on the experimental results and their interpretations and further clarification and discussions are required. Therefore, my opinion is that the manuscript is not significant enough to warrant publication in Nature Communications, as it stands.

I suggest some points for improving manuscript:

1. On page 5, the authors mention that the one- and two-photon activities of the exciton transitions can provide the information about symmetry breaking because their parity selection rules are determined by the crystal symmetry. On the other hand, the authors state that the parity selection rules in the band transitions hold only at high-symmetry points in the Brillouin zone. Because the exciton states consist of the band states with different wave vectors, i.e., the states not at the high-symmetry points, why do the parity selection rules of the exciton transitions simply reflect the crystal symmetry?

Author Response: We thank the Referee for this important question. In band-to-band absorption transitions, the k-selection rule requires that transitions be near vertical (vertical to within the sum of the wave vectors of the photon (or photons) absorbed) in the energy versus wave vector **k** band-structure diagram. Of course, as the referee notes, in this picture, band-to-band transitions can occur from any point in k-space subject to the requirement of energy conservation. As a result, the selection rules for a particular near-vertical band-to-band transition at a point **k** are set by the symmetry of the particular point in k-space where the transition occurs (the so-called “little group” of **k**), not the symmetry of the point group of the crystal.

The k-selection rule as applied to creation of excitons has somewhat different implications, however: For absorption resulting in the creation of an exciton, the initial state is the crystal ground state (full valence band, empty conduction band) with wave vector **K**=0, while the final state is an exciton with center-of mass wave vector **K**, which by momentum conservation must be equal to the photon wave vector in single photon absorption or the sum of the photon wave vectors in the case of two-photon absorption. The wave

vector of a photon is at most of order 0.01 nm^{-1} which is of order $1/1000^{\text{th}}$ of the scale of the Brillouin zone boundary. Consequently, the exciton symmetry is approximately that at the zone center, $\mathbf{K}=0$, (the Γ point) which is the point symmetry of the crystal.

2. On page 6, the authors mention that the doubly-peaked TPA spectrum provides a signature of Rashba splitting. However, I think that a double peak spectrum can be also obtained if the higher energy band exists. How do the authors distinguish these?

Author Response: The transitions into higher lying bands occur at much higher energy in MAPbBr₃. We have noted the response due to a higher lying band at 2.2 eV in MAPbI₃. We have determined the bands energies experimentally by Electroabsorption (EA) spectroscopy, as presented in the Supplementary Figure S13 of the revised SI (originally, Fig. S12). The EA spectrum shows that the second CB in MAPbBr₃ occurs near or above 3.3 eV. The energetic position of the second CB does not change much with temperature since it is determined by the spin-orbit coupling and is therefore weakly dependent on the crystal phase (see for instance: J. Even et al. J. Phys. Chem. Lett. **4**, 2999-3005 (2013)). We have added the following discussion on page 12 of the revised manuscript in order to clarify this point:

“Other possibilities for the appearance of an additional TPA band at this photon energy can be ruled out. First, the higher lying conduction bands occur at much higher energy. Electroabsorption measurements at room-temperature place the 2nd conduction band (CB2) of MAPbBr₃ at or above 3.3 eV. Meanwhile, the energetic position of CB2 does not change much with temperature since it is determined by the spin-orbit coupling and is therefore weakly dependent on the crystal phase⁵¹.”

3. Why are the absolute values of β for MAPbBr₃ at room temperature different between Fig. 3b and Fig. 4a? This is a serious issue regarding the credibility of experimental data and analytical models. The absolute values near the bandgap in Fig. 4a are consistent with the recent report for MAPbBr₃. Therefore, I doubt that the absolute values of analytical models are wrong.

Author Response: The values in Fig. 3b are the results of theoretical calculations, whereas the values in Fig. 4a are experimentally determined values via intensity dependent transmission. As the reviewer has noted, our experimental values are consistent with other reports of the two-photon absorption coefficient found in MAPbBr₃ crystals including [G. Walters et al., ACS Nano **9**, 9340 (2015); R. A. Ganeev et al., Opt. Mater Express **8**, 1472 (2018); and C. Kriso et al. Opt. Lett. **8**, 2431 (2020)]. We have clarified this point by adding the following references and the following sentence to the bottom of page 12 of the revised manuscript:

“The measured values at room temperature are consistent with those previously reported⁴⁷⁻⁴⁹.”

4. In Fig.3 c and d, why does the non-monotonic structure, i.e., peaks around 1.6, 1.85, and 2.05 eV, appear in the TP-PLE data below 2.2 eV, which cannot be seen in the theoretical result? Does it reflect the Rashba splitting?

Author Response: The Referee’s acute eye has observed an additional contribution to the two-photon photoluminescence excitation spectra found in MAPbI₃. During the course of our additional measurements designed to address the comments from Reviewer #2, we observed a more accurate TP-

PLE spectrum from MAPbI₃ crystal due to higher intensity of the pump. The additional peaks are much more pronounced in the resulting spectrum and can be attributed to the exciton contribution near the band gap and the separation of transitions from the Rashba split valence band and conduction bands. These spectra can be seen in Fig. 3c,d for MAPbI₃ crystal at RT and in Fig. 4d for MAPbI₃ crystal at low-temperature.

5. In Fig.4a, I doubt the feasibility of the assignment of the additional band on top of the usual interband contribution to the Rashba band structure. Because the trap state and polaron transitions have similar energy scale to the Rashba splitting [ACS Materials Lett., 2, 20–27 (2020)], do the authors comment on that and confirm that the Rashba splitting is the main origin by, for example, performing the polarization-selective spectroscopy?

Author Response: We note that the contribution of trap states and polarons would give rise to absorption features below the gap that are not observed. Meanwhile, trap-state-enhanced two-photon absorption would require trap states that are near half-gap, while theoretical predictions of trap state energies find these levels sit vary near the conduction band minimum. (see for example T. Shi et al. Appl. Phys. Lett. **106**, 103902 (2015).). We have clarified this point by continuing the discussion added on page 12 of the revised manuscript described in our response to comment 2 above:

“Furthermore, the presence of trap state or polaronic absorption would give rise to features below the band gap that are not experimentally observed, and thus cannot explain the additional feature above the band gap. Meanwhile, trap-state enhanced TPA would require states that lie near but slightly above the mid-gap, while theoretical predictions place such levels vary near the conduction band minimum⁵².”

6. Based on Fig. S13, the authors claim that the bandgap blueshifts at low temperatures. However, this is not acceptable. It is well known that the bandgap of MAPbBr₃ redshifts at low temperature. Therefore, the author's claim that the enhancement of TPA cannot be explained by the traditional band-gap scaling law is also incorrect. The authors should consider that the TP-PL spectrum for perovskite bulk crystal is strongly affected by the reabsorption effect.

Author Response: It is true that the bandgap of perovskites tends to redshift with decreasing temperature but it also blueshifts at the phase transitions. The small blue-shift between emission at room temperature and 50K is consistent with other reports such as H. Linnenbank et al. Opt. Mater. Express 8, 511 (2018). Furthermore, the data shown in Fig. S13 (Fig. S14 of the revised S.I.) compare the two-photon absorption of the same crystal at the two temperatures and therefore the reabsorption effect is comparable in both measurements.

7. According to Fig. S13, the linewidth of TP-PL at 50 K is too large (about 70 meV). This suggests that the free exciton emission was strongly reabsorbed and the observed spectrum was dominated by the bound exciton emission. The author should consider the effect of bound exciton on Rashba splitting.

Author Response: The Referee raises an interesting point about the nature of the emitting excitonic state. Because photoluminescence originates from the lowest energy state following relaxation processes,

the PL may indeed come from the bound exciton at low temperature as the Referee has pointed out. However, the two-photon absorption process occurs at energies significantly higher than the emission and represents the absorption into the conduction bands. The strength of the emission thus depends on the number of absorbed photons regardless of the nature of the emitting state and the conclusions of this work are unaffected by whether the emission comes from either the free or bound exciton.

8. On page 9, the authors mention that MAPbBr₃ at 50 K exhibits the Rashba signature. Is it able to mention whether the main contribution stems from surface or bulk?

Author Response: The large penetration depth of the exciting photons under two-photon excitation conditions guarantee that the bulk is probed. The surface contributions are negligible here. We also note that the one-photon PLE is essentially featureless and mainly decreases with increasing photon energy due to larger surface recombination with reduced photon penetration depth.

We have made these points explicit on page 7 by adding the following statements and by adding the one-photon PLE spectra in Supplementary section S5 as Figure S11.

“For two-photon absorption, the penetration depth is determined by the inverse of the product of the pump intensity and two-photon absorption coefficient β . In our case, the penetration depth is of the order of 1 mm.”

“We further note that the one-photon PLE spectra of these compounds are featureless, exhibiting only a decrease in PL with increasing photon energy due to enhanced nonradiative surface recombination (Fig. S11).”

9. The experimental conditions described in the main text and the materials & methods are different (the laser pulse width and spectral range).

Author Response: We thank the Referee for pointing out this error. The description in the main text reflects the relevant spectral range for the experiments reported in the manuscript, whereas that in the methods section describes the capability of the experimental system more generally. We have adjusted the broader spectral range described in the Methods section to be consistent with the description in the main text. The erroneous pulse width in the methods section was a typo and has also been corrected.

10. The authors say that the TPA spectra β were obtained by integrating the TP-PL spectra from each excitation energy and normalized by the square of the laser power I_0 . However, the obtained spectra should also multiply the excitation photon energy $\hbar\omega$, because the PL intensity is proportional to the excitation carrier density, which equals to $\beta I_0^2 \Delta t / 2\hbar\omega$ where Δt is the excitation pulse length. In addition, the carrier density profile also depends on the value of βI_0 (see, for example, [phys. stat. sol. (b) 245, 2676–2679 (2008)]). The equation in this reference also suggests that if the value of βI_0 is too large, the spectral shape of TP-PL should be changed because of the modification of reabsorption effect. Therefore, the authors should check whether the spectral shapes of TP-PL for each excitation energies were unchanged or not.

Author Response: We thank the Referee for correctly pointing out that the photon energy need be included in the normalization of the TP-PLE spectra in addition to the intensity of the pump. Since the excitation pulse length Δt is the same for all pump energies we have renormalized all TP-PLE spectra by multiplying the spectra by $2\hbar\omega$ in accordance with correct procedure. We note that the overall effect is not very significant because the photon energies vary only by a factor of about 1.5 across the spectra. We have also corrected the description of the normalization procedure in the Methods section.

We also provide below the TP-PL spectra vs excitation energy in order to demonstrate that the spectral shape is unchanged.

Figure R1. Normalized TP-PL spectra used to calculate the TP-PLE spectra for: **a**, MAPbBr₃ at room temperature; **b**, MAPbI₃ at room temperature; **c**, MAPbBr₃ at 50K; and **d**, (PEA)₂PbI₄.

11. Is it possible to discuss the contributions of biexcitons or defect states to the observed data?

Author Response: We thank the Referee for this interesting question. TP-PL directly probes the absorption into excited states that are allowed by symmetry; for example, if the inversion is part of the symmetry group then the TP-PLE would show only states with even symmetry. If the biexciton states or impurity states are allowed by symmetry we would expect that TP-PLE would unravel these ‘below-the-exciton’ states. However, in this case the TP-PL emission spectrum would change. We have not observed a change in the emission spectrum in our measurements.

Reviewer #2 (Remarks to the Author):

In the paper entitled "Broken Inversion Symmetry and Rashba Splitting in Organic-Inorganic Lead-Halide Perovskites Revealed through Two-Photon Absorption Spectroscopy" by Lafalce et al., the authors report an all-optical method for observing Rashba band splitting in several lead-based hybrid perovskites based on two-photon absorption. Using the technique, they claim to observe signatures

of bulk Rashba band splitting in methylammonium lead bromide at low temperatures and phenyl ethyl ammonium lead iodide at room temperature. Furthermore, they observe an absence of the Rashba effect in methylammonium lead bromide and methylammonium lead iodide at room temperature.

The hybrid lead-halide perovskites have attracted much attention in recent years for their superior photovoltaic properties. It was initially suggested that broken inversion symmetry (through the associated bulk Rashba effect) could explain the long carrier lifetimes of this class of materials. Since that conjecture, the question of whether these materials maintain or break inversion symmetry has been a controversial issue. I believe this paper could be of great interest to the community by shedding some much-needed light on the problem. I largely find the technique, data, and analysis presented in the paper to be convincing, but I have a few questions that should be addressed by the authors:

(1) Is Rashba splitting observed in methylammonium lead iodide at low temperatures? If the results of this work are to be accepted, an inversion-symmetry-breaking phase transition occurs in the bromide material. It would be interesting to determine if that is a general phenomenon of the hybrid perovskites, or peculiar to this particular member of the family.

Author Response: The Referee asks an important question about a more comprehensive picture of the bulk Rashba effect in the halide perovskites. While our previous attempts to obtain the TP-PLE spectrum from MAPbI₃ at low temperature were unsuccessful, we have since succeeded in performing these measurements by modifying our experimental apparatus to include a liquid nitrogen cryostat in close proximity to our pump source. This modification also has allowed for higher intensity excitation of the sample at room-temperature and subsequently a more accurate determination of both the room- and low-temperature TP-PLE spectra. These additional spectra can be seen in Fig. 3c,d for MAPbI₃ crystal at RT and in Fig. 4d for MAPbI₃ crystal at low-temperature. We find evidence of Rashba splitting in this material independent of temperature and crystal phase.

The occurrence of the Rashba splitting was also determined to be correlated with greater than quadratic pump intensity particularly near the band edge as presented in Fig S15 in the revised Supplementary Information. The greater-than-quadratic pump intensity dependence can be attributed to a greater-than-linear dependence of emission intensity per photon as was observed in one-photon PL measurements in other studies [Phoung et al. J. Phys Chem. Lett. 7, 2316-2321 (2016)]. Theoretical analysis shows that such pump intensity dependence of photoluminescence occurs when the emitting species involves excitons that bind to donor or acceptor sites [Schmidt, T et al. Phys. Rev. B 45, 8989-8994 (1992)]. The donor/acceptor sites, in turn imply defects and vacancies and a concomitant distortion of octahedral sites that may give rise to a local Rashba effect.

The origin of the Rashba effect and the role of defect sites in perovskite crystals is further demonstrated in the observation of the effect in *mixed halide crystals* as discussed in response to the third comment from Reviewer #3 and presented as Fig. 5a in the revised manuscript.

(2) The authors claim that their technique is bulk-sensitive. This is a critical aspect of the experiment because the Rashba effect will always exist at surfaces. What is the actual penetration depth of the photons used here, and how has it been determined? Is there any other evidence for the distinction of bulk from surface effects with this technique?

Author Response: We thank the Referee for this important question. The penetration depth is given by the inverse product of the two-photon absorption coefficient and the pump intensity and is of the order of 1 mm, which is comparable to the thickness of the crystal. The difference between bulk and surface is evident in the difference between the one-photon PLE and TP-PLE, where the former is featureless except for a decrease in PL with increasing photon energy due to the enhanced nonradiative surface recombination into surface trap states.

We have made these points explicit on page 7 of the revised manuscript by adding the following statements, and also adding the one-photon PLE spectra in Supplementary section S5 as Figure S11.

“For two-photon absorption, the penetration depth is determined by the inverse of the product of the pump intensity and two-photon absorption coefficient β . In our case, the penetration depth is of the order of 1 mm.”

“We further note that the one-photon PLE spectra of these compounds are featureless, exhibiting only a decrease in PL with increasing photon energy due to enhanced nonradiative surface recombination (Fig. S11).”

(3) What is the explanation for the apparent change in the power law exponent between low and high power in the green and black curves in Fig. 2b? More specifically, the solid curves are not straight lines (as claimed in the figure caption), and so do not actually represent any power law. I think it would be less confusing if only straight line fits (i.e. true power laws) were plotted, even if they only match the low power data points.

Author Response: The Referee makes a valid point about the change in the TP-PL power law with intensity. The decrease in the power exponent at high intensity signifies the reduction in PL efficiency with increasing pump intensity and is due to the onset of ‘bimolecular’ recombination that scales as the square-root of the generation rate and possibly further reduced by the onset of pump-induced sample heating effects that increase the nonradiative recombination rate. We have taken the Referee’s advice and adjusted Fig. 2b to only fit the power law below the onset of saturation.

(4) Line 170: Should be "Fig. 2b", not "Fig. 1b".

Author Response: We thank the Referee for pointing out this typo and have corrected it.

(5) The authors should comment on whether their results address the static Rashba effect or the dynamic Rashba effect (i.e. what time scales is this technique sensitive to?).

Author Response: The Referee brings up an important point about whether the observed effect results from static or dynamic inversion symmetry breaking. The 7 ns pulses used to excite TP-PL in this work are several orders of magnitude longer than the timescale of transient structural distortions responsible for the proposed dynamic Rashba effect. Furthermore, it has been argued that the dynamic effect cannot contribute to significant energetic splitting in thermal equilibrium for a centrosymmetric crystal [M. Schlipf and F. Giustino arXiv:2004.10477 (2020)]. Lastly, we believe the discussion on pgs. 13 - 14 of the revised manuscript regarding the spectrally nonuniform intensity dependence displayed in Fig. S15 and the TP-PLE of mixed halide crystals in Fig. 5a make clear that we believe the origin of symmetry breaking results from the presence of interstitial anions producing a static electric field.

The following suggestions would help place this work in broader context:

(6) The authors should address if and how their work can potentially explain the sometimes contradictory/inconsistent results in the literature. For example, Ref. [25] claims inversion asymmetry in the bromide material, while Ref. [26] claims the opposite in the iodide.

Author Response: We thank the Referee for raising this important issue. With the new results that have been obtained for the revised manuscript we believe we can now adequately address this question. Since the inversion symmetry breaking that is necessary for the occurrence of the Rashba effect is shown to result from the presence of ionic defects, the contradictory claims in other reports may be the result of different fabrication procedures leading to different concentrations or spatial inhomogeneity of such ionic defects. We have specifically addressed this point on page 15 of the revised manuscript by adding the following statement:

“Furthermore, we point out that our determination that the origin of symmetry breaking results from ionic defects may help to explain the inconsistent reports about the presence or absence of Rashba splitting in HOIP, since the concentration of these defects may vary depending on the fabrication procedures or exhibit spatial inhomogeneity.”

(7) The authors should emphasize the (important!) consequences of their result. If there is no room-temperature Rashba effect, as they claim in methylammonium lead bromide and iodide, then the theory that it could give rise to the enhanced carrier lifetimes observed in these materials [see for example Refs. [23], [24], or APL Mater. 4, 091501 (2016)] is falsified, and another explanation is needed.

Author Response: We appreciate the Referee’s suggestion to relate the conclusions of this work to the important consideration of long carrier lifetimes in these materials. Since our improved understanding shows that Rashba splitting occurs in the bulk of these materials as a result of ionic defects, we have added the following statement to the concluding paragraph on page 15 and added the suggested reference in the statement that precedes it as reference 56:

“Our observations show the importance of understanding and controlling defect physics for further development of optoelectronic and spintronic applications based on the hybrid perovskites. Particularly, the ability to controllably dope HOIP with optimal defect concentration may be the key to both achieving long, stable carrier lifetimes in photovoltaic devices and for utilizing the Rashba effect in spintronic devices.”

Reviewer #3 (Remarks to the Author):

The manuscript describes two-photon absorption measurements on three halide perovskites MAPbBr₃, MAPbI₃ and the layered materials (PEA)₂PbI₄ presented as evidence for Rashba splitting in those materials. The amount of experimental and theoretical work presented is impressive and extremely interesting, I have, however few comments and questions for the authors before considering publication in Nature Communications:

- I had a doubt reading the manuscript: According to the authors is there or is there not a Rashba signature for MAPbBr₃ at room-temperature (Fig. 3)? It seems they are defending an absence of splitting for MAPbBr₃, but not for MAPbI₃. Is that correct? There should be a very clear sentence in the manuscript to conclude this part.

Author Response: We appreciate the Referee's concern and the need for clarification on this point. Our results suggest that the Rashba effect is not observable for MAPbBr₃ at room-temperature but there is evidence for Rashba splitting in MAPbBr₃ at low temperature, as well as for MAPbI₃ at room temperature and low temperature, in mixed-halide crystals and for (PEA)₂PbI₄. To make certain these conclusions are clearly presented in the manuscript we have added to the introduction on pages 3 – 4 the following paragraph:

“Here we use two-photon absorption (TPA) spectroscopy to clarify the role of inversion symmetry breaking and its relation to the Rashba effect *in the bulk* of HOIP. We show that inversion symmetry breaking and Rashba-splitting is correlated with the presence of ionic defects in the crystals and can thus be attributed to local electric fields. Specifically, we demonstrate Rashba splitting in single crystals of MAPbI₃ at ambient and cryogenic temperature, and show that Rashba splitting is induced by defects that occur in MAPbBr₃ during the transition to the low-temperature orthorhombic phase. Additionally, we observe the signature of the Rashba effect in single crystals of MAPbI_xBr_{3-x} and two-dimensional (2D) perovskite phenyl ethyl ammonium lead iodide (PEA)₂PbI₄ due to intrinsic inversion symmetry breaking that results from halide alloying and 2D layered structure, respectively.”

Additionally, we revised the concluding paragraph on page 15 to include the following sentence:

“We show that inversion symmetry is broken in 3D perovskites by local electric fields due to ionic defects. These defects are present naturally in our MAPbI₃ crystals and can be directly engineered by alloying halide anions in MAPbI_xBr_{3-x} crystals. On the other hand, defects are induced upon structural distortion during the temperature-driven phase transition as in low temperature MAPbBr₃. The mechanism we have elucidated occurs in addition to the natural inversion symmetry breaking that occurs in the layered 2D perovskite (PEA)₂PbI₄.”

- The interpretation of the spectra as being evidence of Rashba coupling, relies on the fitting of the experimental data by a model. Sadly, this model is entirely placed in the Supplementary Information

document, making the main text not readable by itself. The model should absolutely be discussed in the main text.

Author Response: We have taken the reviewer's suggestion and included the discussion of the k.p model including the Rashba interaction in the main text on pages 10-11 of the revised manuscript.

- The general consensus is that the low-temperature phase for MAPbX₃ belong to the Pnma group, i.e. a group that presents symmetry of inversion, hence no possible Rashba coupling. However, you find close to record values for the coupling in those materials. This should be discussed in details! Is the Pnma group not a correct choice for those materials? Is there another source of symmetry breaking?

Author Response: The Referee is correct that the consensus in the field is that the low-temperature orthorhombic crystal phase in this class of perovskite materials is the *Pnma* space group, which is inversion symmetric and should preclude the Rashba effect. We do not consider this consensus to be incorrect. We believe the origin of inversion symmetry breaking in the low-temperature phase of MAPbBr₃ results from dislocations of halide ions created by strain during the temperature-induced phase transition, while in MAPbI₃ these defects are present in the as-synthesized crystal. In order to test this hypothesis, we fabricated and examined the TP-PLE spectrum of mixed-halide perovskite crystals with low-iodide content.

We have added the results of this work as Fig. 5a in the main text and the following discussion in the main text on page 14 of the revised manuscript. We have also included the one-photon PL used to identify the exciton positions as Fig. S16 in the Supplemental Information, where we also include comparison of the TP-PLE data to the single band model with exciton enhancement that fit well the spectrum of MAPbBr₃ as Fig. S17 showing that the TP-PLE spectrum of mixed crystals cannot be accounted for by such a model.

“In order to further test this hypothesis, we examined the TP-PLE spectrum of mixed-halide perovskite crystals, MAPbI_xBr_{3-x} with low-iodide content, x. In doing so, we avoid inducing a phase transition from the cubic to tetragonal phase while explicitly breaking local inversion symmetry through the mixing of halide anions of different size. The results are shown in Fig. 5a. In comparison to room temperature MAPbBr₃, the spectra obtained from MAPbBr_{2.7}I_{0.3} and MAPbBr_{2.4}I_{0.6} show distinct features consistent with the method described herein for using two-photon absorption to observe inversion symmetry breaking and the Rashba effect. Firstly, we observe the shoulders near the onset of TP-PLE at 2.18 eV for MAPbBr_{2.4}I_{0.6} and at 2.24 eV MAPbBr_{2.7}I_{0.3} that demonstrate the two-photon allowed nature of the exciton transition via inversion symmetry breaking. The identification of these features as exciton peaks is confirmed by the energetic location which directly coincides with the peak position of the one-photon photoluminescence from these crystals (See Supplementary Figure S16). Secondly, we observe a pronounced feature that occurs above the band-gap consistent with the observation of the transitions between levels of Rashba-split valence and conduction bands. The positions of these bands are found to occur at 0.16 eV and 0.12 eV above the exciton position for MAPbBr_{2.7}I_{0.3} and MAPbBr_{2.4}I_{0.6}, respectively”.

- The shape of the MAPbI₃ spectra has been observed and interpreted differently earlier (Wei et al. Nat. Commun. 10 5342) without invoking Rashba splitting. This should be discussed extensively as it

is unexpected for Pnma materials to show Rashba signatures. You need to make sure no other interpretation of the spectra can be made.

Author Response: We thank the Referee for the important question regarding the comparison of the work by Wei et al. As in that work, we have interpreted the spectra as resulting from the inclusion of contributions of two band-to-band transitions. We have clarified this point by including the reference in the manuscript as reference 44 and pointing out the consistency of the result and interpretation by adding the following statement on page 9 of the revised manuscript:

“Wei et al. also observed this higher energy band using two-photon induced microwave conductivity and similarly attributed it to the contribution from the split-off conduction band⁴⁴.”

REVIEWER COMMENTS

Reviewer #1 (Remarks to the Author):

I confirmed the replies from the authors and the revised manuscript and supporting information. I appreciate the efforts of the authors to answer my questions and comments. The authors have properly addressed most of my concerns. However, I still have some questions and concerns as stated below. I suggest that the following points should be considered.

1. I am still concerned whether the theoretical calculations including the Rashba effect well reproduce the experimental results (Figs. 3d, 4a and 4d). Because the agreement between the theory and the experimental data verifies the interpretation as the Rashba effect, it is important to show the agreement more clearly. On page 11 of the manuscript, the authors state that "the model produces an average of splitting in valence and conduction bands because of assumptions of equivalent effective masses and uniform distribution of the joint density of states". If the effective masses and/or the joint density of states are properly adjusted, can the theoretical calculations well reproduce the peak structures appearing in the experimental results?
2. According to Fig. S11, the one-photon PLE for MAPbI₃ is apparently different from the linear absorption spectrum. The authors mentioned that this is due to the contribution of non-radiative surface recombination. Is it possible to interpret that such contribution caused the significant sharp peak to appear in the two-photon PLE spectrum for MAPbI₃?
3. In the response for comment 6, the authors mentioned that "the reabsorption effect is comparable in both measurement". However, it should be noted that the Urbach energy of the absorption depends on the temperature (for example, Ledinsky et al., JPCL 2019, 10, 1368-1373), and thus the reabsorption effect cannot be comparable between different temperatures especially in the case of two-photon PL for optically thick crystals.
4. At the 3rd line on page 11, there is a sentence "...matrix elements $\langle 4|H_v|3\rangle$ and ...". However, I think this should be "...matrix elements $\langle 4|H_c|3\rangle$ and ...".
5. At the last sentence in the 2nd paragraph on page 12, "...such levels vary near the..." should be "...such levels very near the...".
6. At the last sentence in the 1st paragraph on page 13, "TP-TPE" should be "TP-PLE".

Reviewer #2 (Remarks to the Author):

The authors have satisfactorily addressed all of my questions and concerns in their revised manuscript. I can now recommend publication in Nature Communications.

Reviewer #3 (Remarks to the Author):

I have read with great interest the response letter from the authors as well as the new version of the manuscript. I remain uncertain that the features observed on the TPA spectra can solely be assigned to the Rashba effect. However, all the elements (experimental details, model, references) are provided for the reader to reach his/her own conclusion.

I would have to remaining remarks:

- It is a semantic point, but I would not state that the authors 'show' that the loss of inversion symmetry comes from ionic defects (i.e. halide vacancies). It is a hypothesis, the most likely one, but I see no characterization that supports that statement.
- I strongly suggest the authors to modify Figure 1 and to display the elements used in the model (Ex, Er, etc.).

I think the work greatly contributes to the ongoing discussion around TPA and Rashba effects in halide perovskites and so could find its place in Nature Communications.

REVIEWER COMMENTS

Reviewer #1 (Remarks to the Author):

I confirmed the replies from the authors and the revised manuscript and supporting information. I appreciate the efforts of the authors to answer my questions and comments. The authors have properly addressed most of my concerns. However, I still have some questions and concerns as stated below. I suggest that the following points should be considered.

1. I am still concerned whether the theoretical calculations including the Rashba effect well reproduce the experimental results (Figs. 3d, 4a and 4d). Because the agreement between the theory and the experimental data verifies the interpretation as the Rashba effect, it is important to show the agreement more clearly. On page 11 of the manuscript, the authors state that "the model produces an average of splitting in valence and conduction bands because of assumptions of equivalent effective masses and uniform distribution of the joint density of states". If the effective masses and/or the joint density of states are properly adjusted, can the theoretical calculations well reproduce the peak structures appearing in the experimental results?

Author's Response: We appreciate the Referee's concern about discrepancies between model calculations and the experimental data regarding the double peaked structure observed in our MAPbI₃ experimental spectrum. We note that the resonance features are described in a conceptual way by the formal expressions for $\chi^{(3)}$ as described in Supplementary section 2. As proposed by the referee we have considered an adaptation of effective masses to yield a double peaked structure for better quantitative comparison with the data. The results are shown below. We find much better agreement when the effective masses are adjusted to $m_e = 0.46m$ and $m_h = 0.10m$ where m_e is the conduction band effective mass, m_h is the valence band effective mass, and m is the free electron mass. The fitting results in values for the Rashba splitting energy of $E_R^{c1} = 0.332 \text{ eV}$ and $E_R^v = 0.015 \text{ eV}$, in the conduction band and valence band, respectively. For the transitions with final states residing in the second conduction band, we have used $m_e = 0.40m$ and obtained $E_R^{c2} = 0.240 \text{ eV}$. We note a wide range of parameters may satisfy the requirements and thus we have constrained the calculations to the condition $\alpha_R^c \approx 2\alpha_R^v$.

Figure R1 | Rashba splitting in MAPbI₃ crystal at low temperature observed by TP-PLE spectroscopy. TP-PLE spectrum of MAPbI₃ measured at a temperature of 77K plotted on a semi-log scale. The lines are $\beta(\omega)$ calculated using the standard model for two-photon absorption in semiconductors described in the text (dashed black) and the model including the Rashba effect (solid gray).

In general, we observe that the difference between the broad feature that describes the TPA spectrum of MAPbBr₃ and the double-peaked sharp features that describe the TPA spectrum of MAPbI₃ is whether or not the Rashba splitting energies in the conduction and valence band are comparable to each other. If the energies are not very different, the dominant transitions are the transitions described by the rates W_{31} and W_{42} as defined by Eqs. S10 in the SI, that correspond the transitions from the lower branch of the VB (band 1) to the lower branch of the CB (band 3) and from the upper branch of the VB (band 2) to the upper branch of the CB (band 4), respectively. The final transition energies ($2\hbar\omega$) involved in this case begin at the Dirac point at $\mathbf{k} = 0$ and since the matrix elements of TPA scale as \mathbf{k}^2 , the resulting TPA is broad.

On the other hand, when the Rashba splitting energies in the conduction and valence band are significantly different, the dominant transitions are the transitions described by the rates W_{31} and W_{32} as defined by Eqs. S10 in the SI, corresponding the transitions from the lower branch of the VB (band 1) to the lower branch of the CB (band 3) and from the upper branch of the VB (band 2) to the lower branch of the CB (band 3), respectively. The final transition energies involved in this case begin at finite values of \mathbf{k} determined by the minimum energy difference between the different bands. This causes the onset of these TPA transitions to be abrupt and produce step-function like features in the corresponding spectrum.

Figure R2 | Direct calculation of the two-photon transition rates and resulting dominant contributions. **a** The dominant transitions that contribute to TPA when Rashba energies are comparable represented as the double arrows, between the Rashba-split bands labeled in order of increasing energy. **b**. The two-photon transition rates, W , calculated as a function of the photon energy (determined by the energetic difference between respective bands for the parameters used to derive the model fitting for MAPbBr₃ namely, $m_e = m_h = 0.1m$, $E_R^c = 0.060$ eV and $E_R^v = 0.015$ eV. **c** and **d**. Same as in **a** and **b** but for the parameters used to derive the model fitting for MAPbI₃ namely, $m_e = 0.46m$ and $m_h = 0.1m$, $E_R^c = 0.332$ eV and $E_R^v = 0.015$ eV.

One significant difference from our interpretation presented in the earlier versions of the manuscript is that we can no longer directly attribute the peak energies to 4 times the Rashba splitting energies but must extract these values from comparison to calculations. The calculated relative transition rates, W , and resulting dominant transitions are summarized in the figure below. In order to account for the complete TPL spectrum, we have utilized a linear combination of the standard model represented by black dashed curve in Figure R1, and the sharp features produced by the model including the Rashba effect.

Lastly, we note that considering the effects of the joint density of states (JDOS) is beyond the scope of the $\mathbf{k} \cdot \mathbf{p}$ theory, but likely can also lead to sharper peaks in the TPA spectrum. Below, in Fig. R3, we show the difference between the joint density states in the case of normal degenerate bands (a) and those with Rashba splitting (b). We see that in the case of Rashba-split bands a ring of maximal JDOS occurs at a value of k comparable with the minimal energy difference between the bands and may contribute further to the development of strong peaks in the spectra.

Figure R3 | Calculation of the joint density of states between conduction and valence bands with and without Rashba splitting. a. The joint density of states between conduction and valence bands without Rashba splitting. **b.** The joint density of states between conduction and valence bands with Rashba splitting in the conduction band described by $k_{0c} = 5 \times 10^5 \text{cm}^{-1}$.

We have replaced Figure 4d in the previous manuscript with Figure R1 above and have added the preceding discussion to the Supplementary Information section S3, including Figure R2 as Figure S9.

2. According to Fig. S11, the one-photon PLE for MAPbI3 is apparently different from the linear absorption spectrum. The authors mentioned that this is due to the contribution of non-radiative surface recombination. Is it possible to interpret that such contribution caused the significant sharp peak to appear in the two-photon PLE spectrum for MAPbI3?

Author's Response: We do not believe that such an interpretation is feasible. We have previously discussed the significant difference in photon penetration depth for one-photon and two-photon. This difference accounts for the dominant surface-trap contribution to the one-photon PLE that is absent from the TP-PLE. We note that there is no sharp feature observed in one-photon PLE excitation measurements that occurs at an energy above the band gap that would correspond to the resonance observed in TP-PLE.

3. In the response for comment 6, the authors mentioned that "the reabsorption effect is comparable in both measurement". However, it should be noted that the Urbach energy of the absorption depends on the temperature (for example, Ledinsky et al., JPCL 2019, 10, 1368-1373), and thus the reabsorption effect cannot be comparable between different temperatures especially in the case of two-photon PL for optically thick crystals.

Author's Response: The Referee brings up a valid point about the different line shape of the Urbach tail at different temperatures. We acknowledge that this leads to uncertainties in the estimation of the band-gap from PL at different temperatures. However, we still believe the two-photon photoluminescence spectra in Figs 4a and 4d show a distinct excitonic shoulder due to

inversion symmetry breaking and the resulting relaxation of the exciton selection rules as discussed in the main text. We point out that the band gap energies obtained by comparison of the TP-PLE spectra to the standard model for TPA in semiconductors also reflects a blue-shift at the lower temperatures, namely from 2.25 eV at room temperature to 2.45 eV at 50 K for MAPbBr₃ and from 1.63 eV at room temperature to 1.73 eV at 77 K for MAPbI₃. With regard to the magnitude of β , the band gap of MAPbBr₃ at 50K would have be ~ 1.6 eV in order to account for the factor of 3 increase observed.

We have included the corroborative information about the band-gap values extracted by comparison to the standard model of TPA without the Rashba effect on the page 11.

4. At the 3rd line on page 11, there is a sentence "...matrix elements $\langle 4|H_v|3\rangle$ and ...". However, I think this should be "...matrix elements $\langle 4|H_c|3\rangle$ and ...".

Author's Response: We understand the Referee's confusion on this point. The operator described in the brackets as H_v is the momentum operator as defined in equation S7 and not the Hamiltonian for the valence band. In order to avoid this confusion for the reader we have used a new notation for the operator, P_v , and defined it in the main text on page 11.

5. At the last sentence in the 2nd paragraph on page 12, "...such levels vary near the..." should be "...such levels very near the...".

6. At the last sentence in the 1st paragraph on page 13, "TP-TPE" should be "TP-PLE".

Author's Response: We thank the Referee for a detailed examination of our work and for bringing these errors to our attention. We have corrected the three typos indicated in comments 5 and 6.

Reviewer #2 (Remarks to the Author):

The authors have satisfactorily addressed all of my questions and concerns in their revised manuscript. I can now recommend publication in Nature Communications.

Author's Response: We are happy to hear that the Referee is satisfied with our responses to the questions raised. We acknowledge the significant improvement in the manuscript as a result of the Referee's keen review.

Reviewer #3 (Remarks to the Author):

I have read with great interest the response letter from the authors as well as the new version of the manuscript. I remain uncertain that the features observed on the TPA spectra can solely be assigned

to the Rashba effect. However, all the elements (experimental details, model, references) are provided for the reader to reach his/her own conclusion.

I would have to remaining remarks:

- It is a semantic point, but I would not state that the authors 'show' that the loss of inversion symmetry comes from ionic defects (i.e. halide vacancies). It is a hypothesis, the most likely one, but I see no characterization that supports that statement.

Author's Response: The Referee is correct that we have not demonstrated that the ionic defects are the cause for inversion symmetry breaking by direct experimental evidence. We have replaced the word "show" or similar inferences in the following instances of the 2nd revision of the manuscript:

Page 2: ... **is found to result from ionic impurities that induce local electric fields.** Replaced by: "in homopolar 3D HOIP crystals is believed to result from ionic impurities that induce local electric fields."

Page 15: **We show that inversion symmetry is broken in 3D perovskites by local electric fields due to ionic defects.** Replaced by: "We observed a correlation between inversion symmetry breaking in 3D perovskites with the presence of defects that may produce local electric fields."

Page 15: **Furthermore, we point out that our determination that the origin of symmetry breaking results from ionic defects may help to explain...** Replaced by: "Furthermore, we point out that our hypothesis that the origin of symmetry breaking results from ionic defect may help to explain...".

- I strongly suggest the authors to modify Figure 1 and to display the elements used in the model (Ex, Er, etc.).

Author's Response: We thank the Referee for the useful suggestion and have added the relevant energetic positions to Figure 1.

I think the work greatly contributes to the ongoing discussion around TPA and Rashba effects in halide perovskites and so could find its place in Nature Communications.

REVIEWERS' COMMENTS

Reviewer #1 (Remarks to the Author):

The authors have carefully addressed all of my concerns, and now I can recommend publication in Nature Communications.

Lastly, I found one typo, so please correct it as follows: The author name of Ref. 58, "Ochara" should be "Ohara".